# In Vitro Conversion of *Coffea* spp. Somatic Embryos in SETIS™ Bioreactor System

**DOI:** 10.3390/plants12173055

**Published:** 2023-08-25

**Authors:** Hugo A. Méndez-Hernández, Rosa M. Galaz-Ávalos, Ana O. Quintana-Escobar, Rodolfo Pech-Hoil, Ana M. Collí-Rodríguez, Itzamná Q. Salas-Peraza, Víctor M. Loyola-Vargas

**Affiliations:** 1Plant Biochemistry and Molecular Biology Unit, Yucatan Scientific Research Center, Street 43, No.130 x 32 y 34, Mérida 97205, Yucatán, Mexico; hugo.mendez@cicy.mx (H.A.M.-H.); gaar@cicy.mx (R.M.G.-Á.); ana.quintana@estudiantes.cicy.mx (A.O.Q.-E.); monodopos@gmail.com (R.P.-H.); 2Yucatan Science and Technology Park, Carretera Sierra Papacal—Chuburna Puerto, Km. 5.5, Sierra Papacal 97302, Yucatán, Mexico; ana.colli@cicy.mx (A.M.C.-R.); itzamna.salas@cicy.mx (I.Q.S.-P.)

**Keywords:** acclimatization, bioreactor, *Coffea arabica*, *Coffea canephora*, micropropagation, scale up, somatic embryogenesis

## Abstract

Somatic embryogenesis (SE) is an excellent example of mass plant propagation. Due to its genetic variability and low somaclonal variation, coffee SE has become a model for in vitro propagation of woody species, as well as for large-scale production of vigorous plants that are advantageous to modern agriculture. The success of the large-scale propagation of an embryogenic system is dependent on the development, optimization, and transfer of complementary system technologies. In this study, two successful SE systems were combined with a SETIS™ bioreactor immersion system to develop an efficient and cost-effective approach for the in vitro development of somatic embryos of *Coffea* spp. This study used an efficient protocol for obtaining somatic embryos, utilizing direct and indirect SE for both *C. canephora* and *C. arabica*. Embryos in the cotyledonary stage were deposited in a bioreactor to complete their stage of development from embryo to plant with minimal manipulation. Following ten weeks of cultivation in the bioreactor, complete and vigorous plants were obtained. Different parameters such as fresh weight, length, number of leaves, and root length, as well as stomatal index and relative water content, were recorded. In addition, the survival rate and ex vitro development of plantlets during acclimatization was assessed. The best substrate combination was garden soil (GS), peat moss (PM), and agrolite (A) in a 1:1:0.5 ratio, in which the bioreactor-regenerated plants showed an acclimatization rate greater than 90%. This is the first report on the use of SETIS™ bioreactors for the in vitro development of somatic embryos in *Coffea* spp., providing a technology that could be utilized for the commercial in vitro propagation of coffee plants. A link between research and innovation is necessary to establish means of communication that facilitate technology transfer. This protocol can serve as a basis for the generation and scaling of different species of agroeconomic importance. However, other bottlenecks in the production chains and the field must be addressed.

## 1. Introduction

Coffee is one of the most significant agricultural products in the world. More than 11 million hectares are cultivated annually, primarily in the tropical and subtropical regions of Africa, Asia, and Latin America (http://coffee-genome.org/; accessed on 12 May 2023), providing work for more than 90 million people. Mexico is a producer of both high-quality and organic coffee. All coffee species are woody perennial trees or shrubs that vary in morphology, size, and ecological adaptation [1]. Although there are more than 124 species in the *Coffea* genus, only two are employed for commercial coffee production: *Coffea arabica* and *C. canephora*, which account for 60% and 40%, respectively, of global production.

*C. arabica* is an autogamous tetraploid species (2n = 4x = 44) formed via hybridization between *C. canephora* and *C. eugenioides*, composed of populations of highly homozygous low-caffeine individuals [2,3]. *C. canephora* is a diploid species (2n = 2x = 22) of allogamous reproduction and consists of polymorphic populations of heterozygous individuals with high caffeine content [1]. The generation of new coffee varieties with enhanced and stable features through horticultural propagation procedures would require around 30 years. Somatic embryogenesis (SE) is one of the most viable biotechnological methods for its enhancement [4].

SE has been presented as an alternate method for propagating coffee cultivars since its earliest studies [5,6,7]. SE is a clear example of propagation based on the totipotency of plant cells, which provides an alternative to standard propagation methods such as seeds, cuts, and grafts. It is also a useful method for creating genetically identical plants [8]. Diverse research groups have lately developed various procedures for obtaining a highly effective, reproducible, and reliable SE process for *Coffea* spp. [9,10,11]. For instance, successful protocols for mass propagation of somatic embryos (56 days after induction) that originate directly from explants of leaf tissue without requiring the proliferation of embryogenic callus have been published [9]. Recently, the use of SE in coffee as a model system in several molecular studies has been described [12,13,14,15,16].

On the other hand, in the 1990s, significant progress began to be made towards commercial scaling of the coffee tree by cultivating embryogenic tissues in a liquid culture medium in Erlenmeyer flasks or a bioreactor, allowing the propagation of elite coffee clones with high levels of heterosis [17,18,19]. Previously, automated temporary immersion systems [recipient à immersion temporaire automatique (RITA©)] have been a useful tool for the production and germination of somatic embryos of *Coffea* spp. [18].

Temporary immersion systems (TIS) are currently one of the most commonly utilized biotechnological methods for the micropropagation of large-scale cultures of differentiated tissues (embryos, shoots, plantlets, and roots) [20]. These systems are used to broaden or improve in vitro regeneration protocols and are based on alternating cycles of immersion between the plant tissue and the culture media [21,22]. In general, the immersion period in bioreactors is a few minutes, whereas the period of exposure to air is several hours; this allows for a better availability of nutrients and prevents physiological changes and deformities in the culture [23]. In addition, short-term immersion stimulates physiological processes such as photosynthesis, respiration, and stomatal function, which improves the development of the crop during ex vitro acclimatization, a vital stage in the micropropagation processes [20]. In previous studies, complete immersion of embryos or plant organs in the liquid culture medium was reported to result in physiological abnormalities and a loss of plant material due to asphyxiation and hyperhydricity caused by the low oxygen concentration and water potential of the liquid medium [24,25,26].

Depending on the species and the morphology of the differentiated tissues and organs, it is necessary to use bioreactors that can provide an adequate microenvironment to enable the physiological growth and development of the culture [27]. Several models of immersion systems for induction or the development of the SE have been published in recent years, including the RITA^®^ system [28], temporary immersion modular bioreactors (BioMINT™ I and II) [29,30], bioreactor Monobloc Advance Temporal Immersion System (MATIS^®^) [31], Gravity Immersion Bioreactors (BIG) [32], temporary immersion bioreactor (TIB) [33], and the SETIS™ systems (Vervit, https://setis-systems.be/home; accessed on 12 May 2023). This last system allows greater light irradiation, contains a stainless steel mesh (pore size 0.9 mm) that facilitates the manipulation of the culture (somatic embryos), and has a storage capacity for a culture medium of up to 4 L [26]. These properties make this system an interesting option for the propagation of commercial coffee and other species.

In summary, scaling up micropropagation techniques has changed substantially since Takayama and Misawa [34] scaled up begonia. The type of bioreactor to be used varies widely depending on the species to be scaled, the propagation protocol, and the growth conditions of the culture, among many other factors. There is no single system for scaling a particular crop. Therefore, each laboratory must establish its scale-up system, taking into account the many factors involved in the process, including production costs.

A detailed description of most scale-up systems can be found in Valdini et al. [35], while Mirzabe et al. [36] published a comprehensive review of temporary immersion systems (TIS). Aguilar et al. [37] undertook a systematic review of TIS for several species, including *C. arabica*.

In our laboratory, we have developed an efficient method for obtaining somatic embryos directly from *C. canephora* leaf explant [9]. For *C. arabica*, we developed a procedure for indirect somatic embryogenesis to obtain the embryos.

## 2. Results

### 2.1. Somatic Embyrogenesis in C. arabica

Our laboratory has developed a method for the SE of *C. arabica* (Figure 1). Leaf explants from in vitro cultivated plantlets of *C. arabica* var. Bourbon were utilized for the induction and proliferation of embryogenic calli (Figure 1A). After 45 days in callus induction media in darkness conditions, explants exhibited white and firm callus (Figure 1B). After 90 days, the development of the embryogenic callus observed in the foliar explants (Figure 1C). Each mature cotyledonary somatic embryo was transferred to a development medium plantlet after 180 days for germination and plantlet development.

### 2.2. Somatic Embryogenesis in C. canephora

Our laboratory has an efficient system for the SE of *C. canephora* (Figure 2). Two stages comprise our model: pre-conditioning and induction. For the induction stage, circular explants of the leaf tissue are obtained (Figure 2A). In this embryogenic system, no visible change was observed in the explants in the first hours and days. In contrast, the thickening of the border and the beginning of cell proliferation were observed between days 7 and 14 after induction (Figure 2B,C). Subsequently, between days 21 and 28, an increase in cell proliferation and the beginning of the development of the proembryogenic mass were observed (Figure 2D,E), which would later lead to the development of somatic embryos. Cell division increased at 35 days of induction, and the development of globular embryos was observed, corresponding to the first embryogenic stage (Figure 2F). Seven days later, on day 42, we observed more development of embryogenic structures in the torpedo and cotyledonary stages (Figure 2G). It is worth mentioning that no oxidation of the explants was observed throughout the entire process. Finally, at 56 days (Figure 2H), it was possible to identify embryogenic structures in all stages of development: globular, heart, torpedo, and cotyledonary (Figure 2I–L), with an average of approximately 300 embryos per explant. The somatic embryos were germinated in a culture medium until complete plantlets were obtained.

### 2.3. Establishment of Culture in SETIS™ Bioreactors

For temporary immersion, immersion times of 1 min were established every 6, 12, and 24 h, with a change of culture medium every 30 days. Our protocol used 1 min immersion every 12 h, as it showed the best conversion response from somatic embryos to plantlets. For both species, the culture time inside the bioreactors was ten weeks. The schematic representation of the establishment of the *Coffea* spp. culture in the bioreactors is described in Figure 3 and Figure 4.

### 2.4. Plant Growth and Development in SETIS™

After ten weeks of culture (Figure 5), the plants were harvested to evaluate their growth in the bioreactors, including fresh weight, plant and root length, and the number of leaves in both species. The data of the variables evaluated are shown in Appendix A. Both species grew effectively in the bioreactors, and their conversion to plantlets was very successful. For fresh weight determination, *C. canephora* plants showed the highest fresh weight of 186.16 mg per plant, while *C. arabica* had the lowest weight of 181.06 mg per plant. Regarding the length of the plants, the most significant growth was observed in *C. canephora* plantlets with a length of 2.93 cm at its tallest point, while *C. arabica* plantlets reached 2.77 cm. Regarding the number of leaves, we observed a similar number between 8.7 and 8.1 per plant in *C. canephora* and *C. arabica*, respectively. Root measurement in both species showed a similar root development with a length of 1.34 cm for *C. canephora* and *C. arabica* at 1.46 cm. Although it seems that *C. canephora* responded slightly better than *C. arabica* in the bioreactors, the differences are not statistically significant following all the parameters evaluated (Appendix A). All the plantlets showed good development, vigor, and reduced transplant time.

### 2.5. Evaluation of the Stomatal Index

In vitro seedlings contained a greater number of stomata and epidermal cells (Figure 6, Table 1). *C. arabica* plantlets under in vitro and ex vitro conditions had a similar stomatal index, whereas for *C. canephora* plantlets under ex vitro conditions had a greater stomatal index. The leaves of the in vitro plantlets exhibited a stomatal density greater than 40, while those developed after transplanting the ex vitro plantlets had a reduced stomatal density (less than 14). It is also interesting to note that the stomata of the in vitro plants remained open, while those of the ex vitro plantlets were closed (Figure 6). Furthermore, the relative water content of the plants ranged between 84% and 89%. No significant variation existed between species or culture conditions (Appendix A).

### 2.6. Substrate Tests

The transition stage of plant material within the grow room to greenhouse conditions is very important. In order for plantlets to successfully acclimate and survive, it is crucial that the substrate is chosen correctly. In order to increase the percentage of acclimatized plantlets that survive, different substrate compositions were investigated (Figure 7). Five weeks later, plantlets transplanted into substrates containing a combination of garden soil, peat moss, and agrolite exhibited better acclimatization and greater survival rates than plants acclimated with just garden soil.

The mixture of garden soil, peat moss, and agrolite in proportion (1:1:0.5) proved to be one of the most effective substrate proportions for the acclimation of *Coffea* plantlets grown in bioreactors. On the other hand, the transplanted plantlets in mixtures of garden soil, peat moss, and agrolite with proportions of 1:1:1 and 1:1:0.5, respectively, did not differ significantly (Figure 7). Considering the expense of the inputs for the ex vitro acclimatization of the plantlets, the optimal substrate was a mixture of garden soil, peat moss, and agrolite (1:1:0.5). After five weeks of acclimation in the substrate, we found that the survival of the plants was rich, reaching 91.1% in the substrate GS:PM:A (1:1:0.5). The lower survival rate was seen in the substrate GS:A, reaching 40% (Figure 7).

### 2.7. Ex Vitro Acclimatization of Plantlets

*Coffea* spp. plantlets were transplanted on a large scale using the above obtained information. For the transplant, ten-week-old plantlets grown in bioreactors in vitro were used. Six weeks after ex vitro transplantation, a high rate of acclimation of *Coffea* spp. plantlets was found (Figure 8).

## 3. Discussion

Coffee is one of the world’s most important crops. Using the *Coffea* genus as a model, we have spent several years using tissue culture to comprehend the embryogenic process of somatic cells. We reported an efficient SE system for *C. canephora* (Figure 2) [9], and here we are reporting a SE system for *C. arabica* (Figure 1). This system is a great model for investigating mechanisms that remain bottlenecks in zygotic embryogenesis. The addition of plant growth regulators (PGR) to the culture media [39] is one of the factors involved in the induction of SE in diverse plant species. Auxins are the most commonly employed PGR in the induction of SE in various species, according to Loyola-Vargas et al. [4,16]. Alternatively, the genetic pattern of the species may be associated with the lack or low capability of embryogenic response [40].

Several protocols exist in the literature for the SE of the genus *Coffea*. Nonetheless, distinct outcomes for the same protocol have been observed [4]. The phase of conversion from somatic embryos to plants is one of the main challenges in SE that influences the large-scale scaling of diverse species [41]. It has been revealed that using cell suspensions of embryogenic lines provides a superior synchronous and considerable formation of somatic embryos [42]. However, they have not been exploited commercially, as plants regenerated from cell suspensions are associated with substantial genetic variability and somaclonal variation [43].

Using a liquid culture medium represents an advantage for the micropropagation of many plant species because it provides uniform culture conditions and enhances nutrients absorption [44]. The challenges associated with hyperhydricity are one of the downsides of employing a liquid media [45]. In micropropagation, plant material that has been regenerated by temporary immersion responds better during the acclimation phase [23]. The complementary usage of immersion systems is typically employed for scaling up and optimizing the embryo-to-plantlet conversion stage [23].

Temporary immersion bioreactor technology has been documented for the micro-propagation of a variety of commercial species, including banana [20], sugar cane [46], eucalyptus [47], raspberry [48], and vanilla [26], among others. The immersion period, the volume of the culture medium, and ventilation for the prevention and accumulation of ethylene are some factors involved in the conversion of embryos to plantlets inside the bioreactor [49]. For example, immersions every 6 h for 15 min have been observed to improve somatic embryo germination in *Coffea* spp. [50]. Albarrán et al. [51] also reported that brief immersion times of 1 min resulted in the production of high-quality embryos. Using additional ventilation for one min every six hours during the in vitro propagation of *Gerbera jamessoni* allows for a better development of the culture and the production of shoots without hyperhydricity [52].

In our case, we placed cotyledonary-stage somatic embryos in the SETIS™ bioreactors. Its conversion into plants was very successful (Figure 5). This achievement is likely due to the fact that plant material was submerged for only one min per twelve hours and the plant material was aerated every two hours. It has been observed that the features of the cycle in temporary immersion reactors determine the final behavior of the material [51]. We note that the setting of the dive times is crucial. For example, a 15 min immersion every 6 h is favorable for converting coffee embryos, while 1 min immersion every 24 h stops the conversion and stimulates the production of adventitious shoots [50]. In *C. arabica*, it has been observed that the frequency of short immersions (1 min) in 1-L RITA^®^ bioreactors stimulates the formation and quality of somatic embryos [51].

For instance, Albarrán et al. [51] reported a 75% conversion rate, whereas Etienne-Barry et al. [18] reported a 78% conversion rate. Söndahl et al. [53] reported the conversion of somatic embryos to 50–70% plantlets and 80–95% survival after five months in the greenhouse for *C. canephora*. The same authors report a 50% conversion rate of *C. arabica* somatic embryos to plants. Another group reported that the RITA^®^ bioreactor and temporary root zone immersion bioreactor system converted somatic embryos to plantlets at a rate of 20% and 84%, respectively [54]. In another TIS, the conversion percentage of *Coffea* spp. seedlings was 84% after 45 days of culture [54]. Compared with embryos cultured in a semisolid medium, the synchronization of development and conversion of somatic embryos cultured in bioreactors was higher [18].

The use of temporary immersion is ideal for the regeneration of shoots; however, a single bioreactor does not satisfy the needs of embryogenic cultures since there are previous steps, such as induction, differentiation, and development [55]. Contrary to several observations in the literature suggesting the variability of the population of plants obtained from somatic embryos in a bioreactor, this benefit exists [56].

Since the pioneering work of Söndahl et al. [57,58], the variability of plants obtained through SE has always been a point of discussion. In our case, in *C. arabica* we have not detected variability in the plants obtained using molecular markers [59]. Bobadilla Landey et al. [60] determined only a 0.74% variation among 200,000 plants in the field. After three years, the polymorphism between the mother plants and their descendants was 0–0.003% and 0.07–0.18%, respectively. DNA methylation analysis showed that there are virtually no changes at the epigenetic level and that the slight variation occurring is mainly due to the loss of 1–3 chromosomes.

On the other hand, in vitro plants enhance their photosynthetic functions by transferring to ex vitro conditions in order to ensure their survival in the exterior environment [61]. We observed alterations in the distribution and structure of the stomata in the ex vitro plants (Figure 6) as well as in other species for which a reduction in the stomatal density of successfully acclimatized plants has been documented [62]. Furthermore, under in vitro conditions, stomata are generally open because they are not functional. In ex vitro plants, the closure of the stomata reduce transpiration [63], maintaining the ideal hydric state of the plants.

Under in vitro conditions, plant material is exposed to favorable conditions such as nutrient availability, high relative humidity, and low gas exchange, among others, which all contribute to high proliferation rates [64]. At the time of transplantation, however, in vitro grown plantlets exhibit a low rate of photosynthesis and incomplete autotrophy due to the low content of photosynthetic pigments and little development of chloroplasts [65,66], poorly functional stomata with a circular shape and structure [64], as well as a weak root system and a reduced capacity for the formation of waxy cuticles [67].

The plantlets produced by the germination of the somatic embryos in the bioreactors possessed an abundance of functioning stomata (Figure 6). These observations have also been reported in other cultures in bioreactors, such as *Vanilla planifolia* (Ramírez-Mosqueda, 2021) and *Coloasia esculenta*. This abundance of stomata likely contributed to the robust growth observed after removal from the bioreactors.

In the systems used for micropropagation, up to 90% of the economic cost is related to the labor used, since a subculture of hundreds of embryos and plantlets is required [68,69]. This cost can be reduced by implementing temporary immersion systems, since they not only reduce economic costs but also improve the quality of the plant material, facilitate the change of culture medium, and reduce the use of a gelling agent and culture containers; that is, the conversion time from embryos to plantlets is minimized. In addition, the conversion rate is much higher than that obtained by conventional methods, such as using semisolid medium [9] or Erlenmeyer flasks with liquid medium [23].

On the other hand, there are several differences in published somatic embryogenesis protocols, including ours. The scaling systems employed also demonstrate a big difference. Most of the reports mention the RITA^®^ system. Although it is helpful to see which system is better for scaling, several factors should not be overlooked, such as the efficiency of the SE system itself, among many other factors.

The main factors to consider in the efficiency of a large-scale production system are the quality of the somatic embryos used and the handling to which the somatic embryos are subjected for their introduction into the bioreactors. These factors will make the system more efficient. Doubling production efficiency can mean a 50% decrease in production costs, e.g., in the system described in this investigation, the immersion times of one min every 24 h, instead every 6 or 12 h, produced a reduction in the time which the seedlings reach 8 cm in height, increasing the efficiency of the process and lowering the production costs. Other significant costs are electricity, labor, and transport. These costs vary significantly from place to place and are critical to production costs. However, efficiency is the major component in calculating the cost per plant. Additionally, one factor that must be considered in the price of the plants is their quality. They are generally more vigorous, grow better, and are more productive. These factors must be shown in demonstration plots.

One of the advantages of the SETIS™ system compared to other TIS is that they have a simple configuration with a minimum of parts, and aeration after each immersion cycle. For example, the RITA^®^ systems have a single container where the plant material and the culture medium are deposited. However, when there are prolonged culture cycles, handling and changing the medium in the only container increases the possibility of contamination of the cultures. In the same way, the BioMINT™ I and II systems consist of a single rectangular container that stores the culture medium and plant material separated by a dividing comb. Being a container that is too exposed, the contamination rate is higher. In the case of the SETIS™ system, most of the problems and disadvantages found in other TIS have been analyzed and improved. In this system, the culture medium and the plant material are in separate containers, facilitating the change of the culture medium without manipulating the plant material and reducing the contamination rate. Generally, the initial outlay for a TIS system can be high. Some parts need to be replaced after several autoclave cycles.

Currently, the existing technology has been improved using low-cost containers, controlling parameters that affect the efficiency of the system and the loss of plant material. The cost and components of the SETIS™ system can be viewed online (https://setis-systems.be/products/setis-bioreactor/setis-bioreactor-product; accessed on 12 May 2023).

This is the first study to use an efficient somatic embryogenesis system of *Coffea* spp. in SETIS™ bioreactors to convert somatic embryos into plants. Our findings show that using temporary immersion systems to develop somatic embryos of *Coffea* spp. can serve as a foundation for generating and scaling various species of agroeconomic significance.

## 4. Materials and Methods

### 4.1. Somatic Embryogenesis Induction in C. arabica

The seeds of *C. arabica* var. Bourbon were washed with a detergent solution, followed by a further wash with 70% ethanol for 3 min and commercial bleach [1.25% (*v*/*v*) free sodium hypochlorite] for 20 min with agitation. After washing thrice with sterile distilled water, the embryos were removed from the seeds and placed in Magenta boxes containing 40 mL in a semi-solid germination Murashige-Skoog medium (MS) [70] containing 29.6 µM thiamine-HCl (Sigma, T3902, St. Louis, MO, USA), 550 µM myo-inositol (Sigma, I5125), 158 µM L-cysteine (Sigma, C-8277), 166.48 mM glucose (Sigma, G-5000), 0.041 µM biotin, 0.54 µM 1-naphthaleneacetic acid (NAA), 2.32 µM kinetin (KIN), and 0.285% (*w*/*v*) Gellan gum (PhytoTechnology Laboratories, G434), (medium G) under photoperiod conditions of 16 h light/8 h dark inside a culture room with a temperature of 25 ± 2 °C, equipped with white-light-emitting diode lamps (LED-T8-S00235, 18W-1530 lm) and a light intensity of 150 µmol m^−2^ s^−1^). The medium was adjusted to pH 5.8 and sterilized at 121 °C (1 kg cm^−2^) for 20 min.

In vitro plantlets were grown in a semi-solid pre-conditioning medium MS salts medium [70] for two weeks containing 11.86 µM thiamine-HCl (Sigma, T3902), 550 µM myo-inositol (Sigma, I5125), 158 µM L-cysteine (Sigma, C-8277), 9.72 µM pyridoxine HCl, 16.26 µM nicotinic acid, 87.64 mM sucrose (Sigma, S539), 0.54 µM NAA, 2.32 µM KIN, (medium PC), and 0.285% (*w*/*v*) Gellan gum (PhytoTechnology Laboratories, G434), under photoperiod conditions as described above.

For callus induction, leaf explants of the second and third pair of leaves of the plant were used. Medium (S1) MS [70] was supplemented with 29.6 µM thiamine-HCl (Sigma, T3902), 550 µM myo-inositol (Sigma, I5125), 210 µM L-cysteine (Sigma, C-8277), 87.64 mM sucrose (Sigma, C-8277), 4.5 µM 2,4-dichlorophenoxyacetic acid (2, 4-D), 9.2 µM KIN and 0.285% (*w*/*v*) Gellan gum (PhytoTechnology Laboratories, G434). Adjust the pH of the medium and sterilize as described above. The explants were placed with the adaxial surface in contact with the callus induction medium (S1) in glass flasks, and incubated in the dark at 25 ± 2 °C for seven weeks. The calli were subcultured every four weeks on MS medium (S2); this medium has the same characteristics of the S1 medium but the concentration of the MS salts is halved. This is for the maintenance of the callus.

For induction of somatic embryogenesis, one gram of friable callus was inoculated into 50 mL of S3 liquid medium in 250 mL flasks. This medium (S3) consisted of half-strength MS salts [70] and an increased concentration of potassium nitrate [KNO_3_ (2x)], supplemented with KNO_3_ 28.18 mM (Jalmek, P5525-05), 29.6 µM thiamine-HCl (Sigma, T3902), 550 µM myo-inositol (Sigma, I5125), 210 µM L-cysteine (Sigma, C-8277), 87.64 mM sucrose (Sigma, C-8277), with 0.27 µM NAA, and 2.32 µM KIN, at 25 ± 2 °C [6]. We adjusted the pH of the medium and sterilized as described above. The cell suspensions were grown in a dark culture room at 25 ± 2 °C on a gyratory shaker at 100 rpm. Every two weeks, subcultures were produced by inoculating 1 g of tissue into 50 mL of new media. Somatic embryos began to appear seven weeks after transfer to SE induction medium. Somatic embryos were collected after six months and transferred to PC medium. The plantlets were maintained under photoperiod conditions as described above.

### 4.2. Somatic Embryogenesis Induction in C. canephora

We started with seeds of *C. canephora* Pierre var. Robusta. The seeds were washed as described above. The zygotic embryos were isolated from the seeds and deposited in magenta boxes (ten zygotic embryos in each box) containing the MS salts [70] maintenance medium as described above. The zygotic embryos were cultured for 24 weeks under photoperiod conditions as described above for their development into plantlets.

In vitro plantlets of *C. canephora* were cultivated in a semi-solid MS salts developmental medium [70] supplemented with 29.6 µM thiamine-HCl (Sigma, T3902), 550 µM myo-inositol (Sigma, I5125), 0.15 µM L-cysteine (Sigma, C-8277), 16.24 µM nicotinic acid (Sigma, N4126), 9.72 µM pyridoxine-HCL (Sigma, P9755), 87.64 mM sucrose (Sigma, S539), and 0.285% (*w*/*v*) Gellan gum (PhytoTechnology Laboratories, G434), under photoperiod conditions as described above. Every eight weeks, the medium was replaced until the plants had three to four pairs of leaves.

For the direct induction of SE, plantlets were preconditioned for 14 days in a semi-solid MS medium [70] supplemented with 0.54 µM NAA and 2.32 µM KIN, at the same conditions as described before (Appendix A). After 14 days, the second and third pairs of leaves were selected and sliced into 0.8 cm diameter (*C. canephora* only) segments with a sterile punch. Five explants were placed in each 250 mL flask containing 50 mL of Yasuda liquid medium with the modified nitrogen source [38] and supplemented with 5 µM 6-benzyladenine (BA) (PhytoTechnology Laboratories, B800), adjusted to pH 5.8 [9]. For 56 days, the flasks were incubated in the dark and agitated (55 rpm) at 25 ± 2 °C. After 56 days, the different embryogenic stages (globular, heart, torpedo, and cotyledonary) were observed.

### 4.3. Establishment of Culture in SETIS™ Bioreactors

The SETIS™ bioreactors (Vervit, Zelzate, Belgium) consisted of a 6 L upper container for plant material with a stainless steel mesh (0.9 mm pore size) and a silicone frame, and a 4 L lower container for culture medium. Both containers are connected by a 9 mm diameter translucent silicone hose, a pair of 65 mm diameter filters (membrane hydrophobic-PTFE, 0.22 µm pore size) attached to each air inlet and outlet area to maintain sterilization. The timing was adjusted so that the plant material was immersed in the liquid medium of 1 min every 12 h and aerated every 2 h. In the establishment of the culture, 1 L of MS [70] liquid culture medium supplemented with 29.6 µM thiamine-HCl (Sigma T3902), 550 µM myo-inositol (Sigma, I5125), 0.15 µM L-cysteine (Sigma, C-8277), and 87.64 mM sucrose (Sigma, S539). We adjusted the pH of the medium and sterilized as described above. One hundred and fifty embryos in the cotyledon stage were deposited in each bioreactor. These embryos were selected according to the protocol described above. The bioreactors with the somatic embryos of *Coffea* spp. were maintained under a photoperiod conditions as described above for ten weeks. The plantlets were then harvested to determine their growth characteristics, including height, number of leaves, root length (Appendix A), stomatal index, and relative water content. Appendix A depicts a description of the SETIS™ bioreactor system.

### 4.4. The Relative Water Content

Leaf discs were cut from three replicates and the fresh weight (FW) was recorded. The explants were then immersed for 24 h in distilled water. The surface water was evaporated and the maximum turgor weight (TW) was measured. The leaves discs were placed in an oven at 65 °C for 48 h or until they reached a consistent dry weight (DW). The RWC was calculated using the formula (FW-DW)/(TW-DW) × 100 [71].

### 4.5. Study of the Stomatal Index

The abaxial leaf surface of plantlets (in vitro and ex vitro) of *C. arabica* and *C. canephora* were was coated with transparent nail polish to obtain epidermal impressions. Fully expanded young leaves were selected. The number of stomata (S) and epidermal cells (EC) were determined per mm^2^. The stomatal index (SI) was calculated using the formula SI = [S/(S + EC)] × 100 [72]. The stomatal density was obtained by counting the stomata per visual field of the microscope (40×).

### 4.6. Ex Vitro Acclimatization of Harvested Plants from Bioreactors

To evaluate the effectiveness of the acclimatization stage of *Coffea* spp. plantlets, the plantlets were removed from the bioreactor and placed in a container with clean water to remove excess culture medium after ten weeks of cultivation, when their height was between 2 and 3 cm. The plantlets were then soaked with rooting agent RADIX^®^ 1500, and they were transplanted subsequently into trays with 50 5 × 7 cm cavities containing a mixture of garden soil, peat moss, and agrolite (1:1:0.5 *v*/*v*) as a substrate. The trays were placed inside a dome-shape plastic container to maintain relative humidity in the plants. The prior acclimatization phase was conducted in a culture room under a photoperiod conditions as described above and 55 ± 5% relative humidity. The plants were fertilized once per week with Roottex^®^ (COSMOCEL^®^) at a concentration of 1 g L^−1^. The length of the plants, the number of leaves, and the length of the roots were subsequently evaluated from at least nine plants of each species (Appendix A).

### 4.7. Statistic Analysis

An analysis of variance (ANOVA) of the survival data of the acclimatized plantlets was performed. For growth and development, *t*-test analyses were performed. The data were analyzed by Origin V8 (Data Analysis and Graphing Software). Three biological replicates of each treatment were used. Each repetition is defined as a plastic container with 15 plantlets. The Tukey test determined the significance of the differences between the means. Differences were considered significant at *p* ≤ 0.05.

## 5. Conclusions

In conclusion, a large-scale scaling process for somatic embryos of *Coffea* spp. in a temporary immersion system has been established in the present work. Our findings indicate that the conversion time from somatic embryos to plantlets is approximately three months. In addition, by optimizing the frequency of immersion and with the appropriate culture medium, this system could work in other embryogenic cultures, maximizing the development of somatic embryos and seedlings. The efficiency of plant adaptation to field conditions was achieved, and this methodology might be used for other species improvement systems. Rooting and ex vitro adaptation are crucial steps in the propagation of plants in vitro; a developed root and a suitable culture room allow better acclimatization of the plants. From a future perspective, this technology can serve as an alternative for commercially cultivating of economically important plants.

## Figures and Tables

**Figure 1 plants-12-03055-f001:**
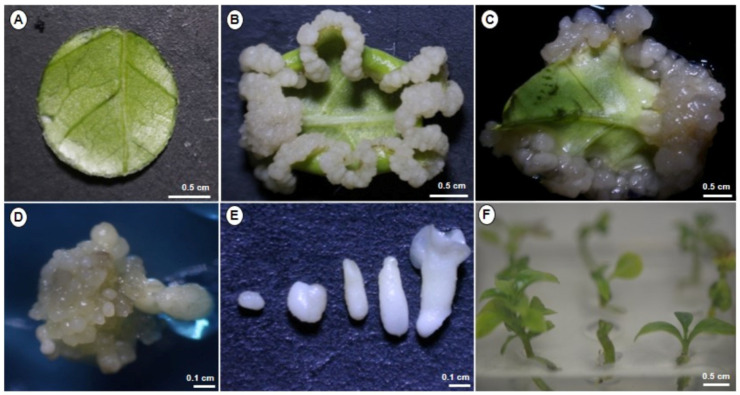
Induction of somatic embryogenesis in *C. arabica*. (**A**) Leaf explant. (**B**) Explant after 45 days of culture in darkness in callus induction medium. (**C**,**D**) Explant showing the embryogenic callus. (**E**) Embryos are in different stages of development (globular, heart, torpedo, and cotyledonary, from left to right). (**F**) Plantlets of *C. arabica* in vitro.

**Figure 2 plants-12-03055-f002:**
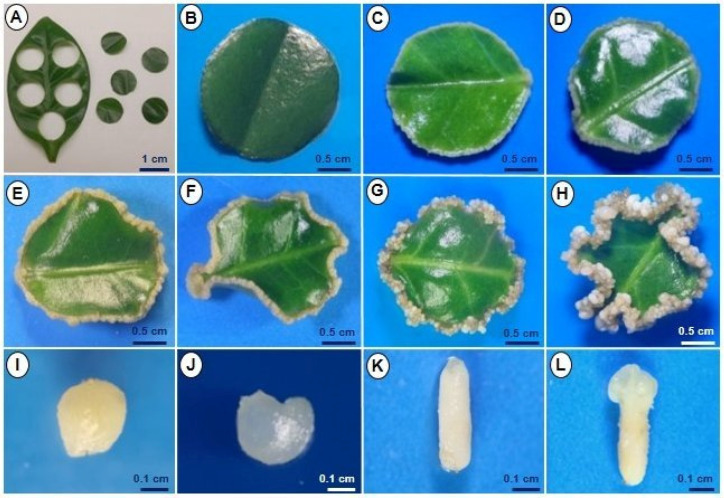
Induction of somatic embryogenesis in *C. canephora*. (**A**) Leaf explants were obtained from plantlets 14 days after the preconditioning stage in MS medium, supplemented with 0.54 μM NAA and 2.32 μM KIN. (**B**–**D**) Circular explants at 7, 14, and 21 days, respectively, after induction of SE in Yasuda [38] medium supplemented with 5 µM BA. (**E**–**H**) Circular explants at 28, 35, 42, and 56 days, respectively, after induction of SE. (**I**–**L**) Embryos at different stages of development (globular, heart, torpedo, and cotyledonary, respectively).

**Figure 3 plants-12-03055-f003:**
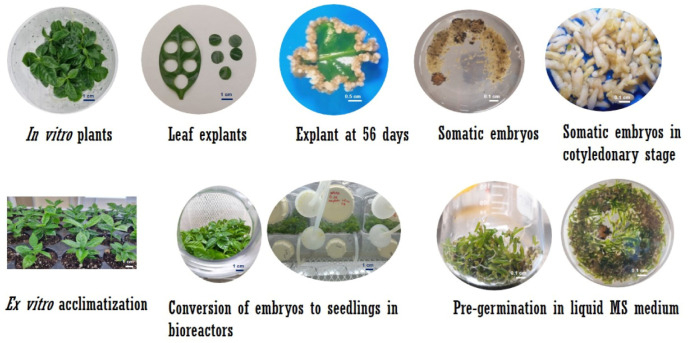
Diagram of the culture of *C. canephora* in temporary immersion bioreactors. As a first step, somatic embryos were obtained as described above. The embryos in the cotyledonary stage were placed in the vessel of the bioreactors for conversion to plantlets. Once the plantlets were obtained, they were placed in trays with the substrate for ex vitro acclimatization.

**Figure 4 plants-12-03055-f004:**
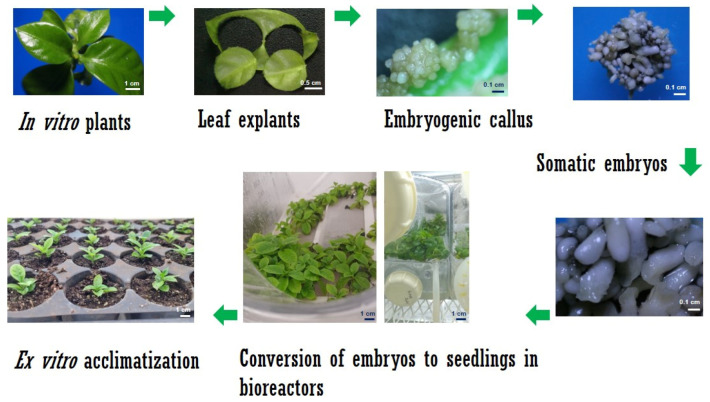
Diagram of the culture of *C. arabica* in temporary immersion bioreactors. As a first step, leaf explants of in vitro plantlets were obtained. After 45 days, the embryogenic callus was obtained, followed by embryos at different stages of development. The embryos in cotyledonary stage were deposited in the container of the bioreactors until their conversion to plants.

**Figure 5 plants-12-03055-f005:**
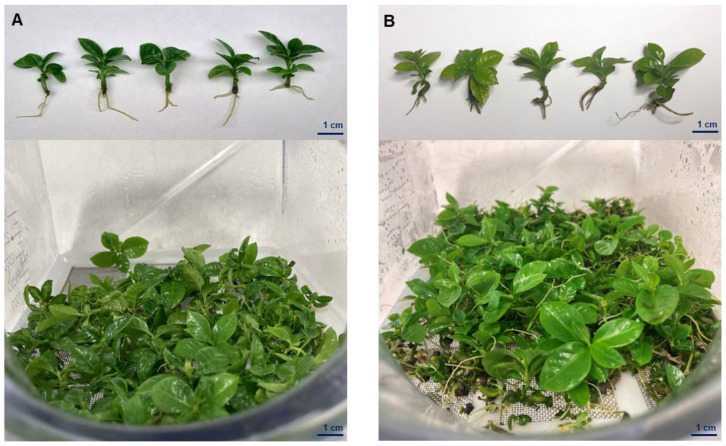
*Coffea* spp. grown in SETISTM temporary immersion bioreactors. (**A**) *C. arabica* and (**B**) *C. canephora* plantlets were evaluated after ten weeks of culture.

**Figure 6 plants-12-03055-f006:**
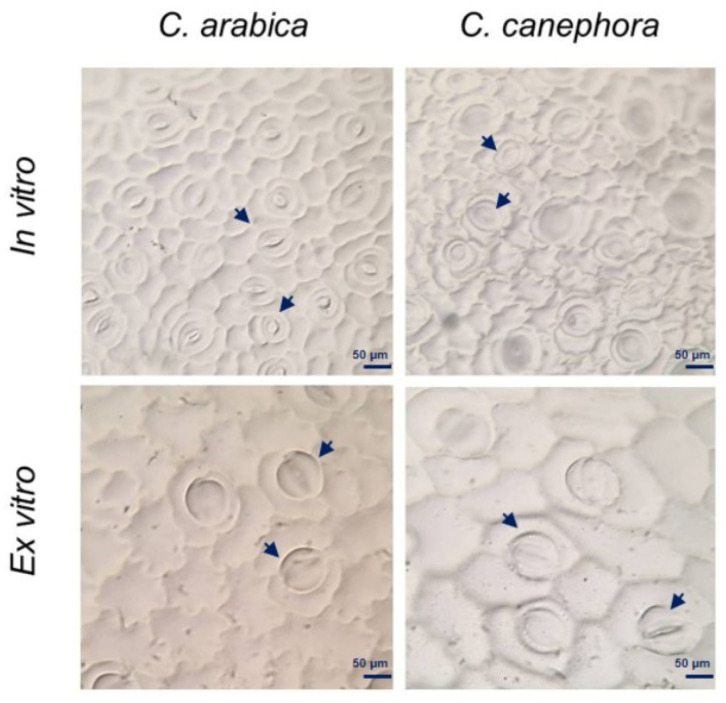
Epidermal impressions of the abaxial surface of the leaves. Presence of stomata and epidermal cells of in vitro and ex vitro plantlets of *C. arabica* and *C. canephora*. Arrows indicate plant stomata.

**Figure 7 plants-12-03055-f007:**
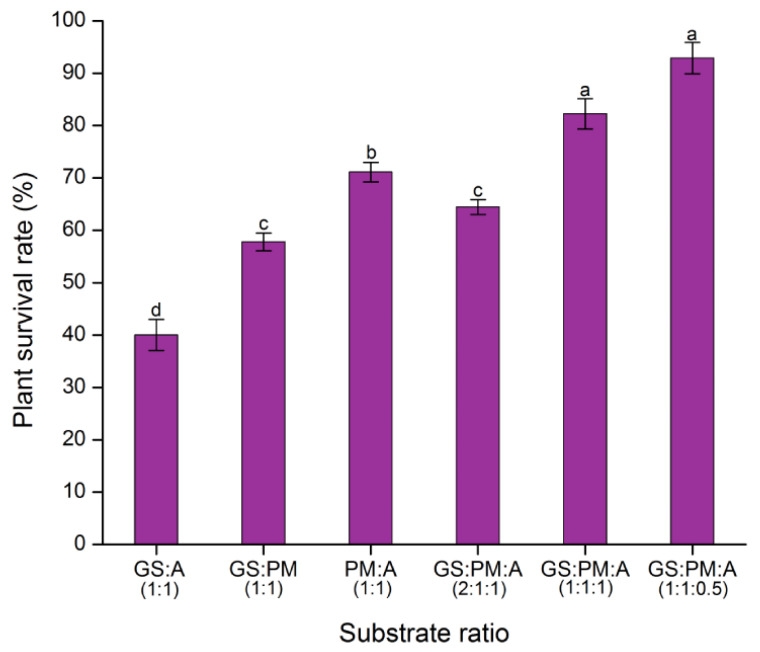
Influence of the substrate on the ex vitro acclimatization of *Coffea* spp. Plantlets were grown in SETISTM bioreactors. Analyzes were performed with three biological replicates of independent experiments (Appendix A). Error bars represent standard error. The letters represent the statistical significance of the mean differences between each determination according to the Tukey test (*p* ≤ 0.05). Abbreviations: GS: Garden soil, A: Agrolite, PM: Peat moss.

**Figure 8 plants-12-03055-f008:**
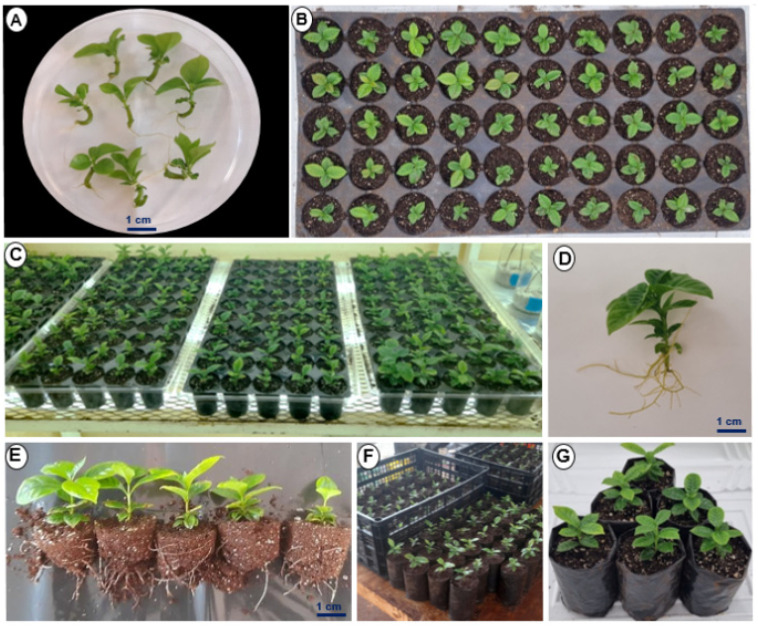
Ex vitro acclimatization of *Coffea* spp. (**A**) The plantlets were harvested from the bioreactors and placed in clean water to remove the excess of medium. (**B**) The base of the plantlets was impregnated with a commercial rooter and transplanted into 50-well plastic trays containing a mixture of GS:PM:A (1:1:0.5). (**C**) The trays were placed inside a plastic container to maintain relative humidity. (**D**) Plant six weeks after acclimatization. (**E**) Plant ten weeks after acclimatization. (**F**,**G**) Transplant from culture tray to nursery bag.

**Table 1 plants-12-03055-t001:** Stomatal index and stomatal density. The number of stomata and epidermal cells was recorded under in vitro and ex vitro conditions of *C. arabica* and *C. canephora* plantlets.

Condition	Species	Stomata	Epidermal Cells	Stomatal Index	Stomatal Density
In vitro	*C. arabica*	7	30	18.3	45.3
	*C. canephora*	5	24	16.2	42.3
Ex vitro	*C. arabica*	2	10	18.8	11.7
	*C. canephora*	2	7	20.6	14.3

## Data Availability

Not applicable.

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
