# Peer review of "In Vitro Conversion of Coffea spp. Somatic Embryos in SETIS™ Bioreactor System"

_plants, 2023, doi:10.3390/plants12173055_

Round 1
Reviewer 1 Report (Previous Reviewer 2)
The revised version is gradually improved, however, the most relevant remarks of the reviewer to the former manuscript are not considered. Therefore, the scientific value is still very low. It is shown, that SETIS works with 2 Coffea species, but the experimental design doesn’t allow to compare the potential of this TIS system with others or at least with methods without TIS-systems. Therefore, the efficacy can't be proved as authors always stress. Another weak point- not improved - is that no quantitative data are presented and evaluated except in table 1 and figure 7.
The description of material and methods need to be improved and text related to material and methods should not be presented in results.
If the title is focusing on conversion, the data presented should be also focused on conversion.
More remarks are annotated to the manuscript.

Author Response
Q: This cannot be an aim of the study! What do you expect from application of SETIS? How SETIS works in comparision with standard procedure or other TIS-systems. Please consider what you discuss and conclude. This bases on the aim and hypothesis.
A: We agree with the reviewer. We have made the changes to the manuscript.
Q: This is material and methods! Do you have quantitative data and statistics? In the title you ention the stage of embryo conversion only, than 2.1 and 2.2 are not necessary in the paper.
A: We agree with the reviewer. However, we consider it necessary to show the induction process of somatic embryogenesis since we start from there as the first step to later move on to bioreactors..
Q: Mark and describe in detail here, as in Fig 2
A: We agree with the reviewer. We have made the changes to the manuscript.
Q: material and method! here only results, and results regarding immersion frequency is welcome! This remark you had already in the first draft.
A: We agree with the reviewer. We have made the changes to the manuscript.
Q: Here are overlapping information with Fig 1. I guess this figure privide sufficient informatuion an Fig 1 is not necessary
A: We appreciate your comment. We consider that Figure 1 is necessary since it describes the process of induction of embryogenesis.
Q: see remarks on Fig 3, I guess Fig 2 is not required.
A: We appreciate your comment. We consider that Figure 2 is necessary since it describes the process of induction of embryogenesis.
Q: Are all these differences significant? Give quantiative data and statistics, pictures are not sufficient. For evaluation of efficiacy of this TIS-system, at the same time ther systems should be evaluated or a standard method without TIS.
A: We appreciate your comment. The statistical analyzes of the growth and development of the plants were carried out, however there was no significant difference.
Q: Are these leaves new, developed after transfer or are these leaves modified in vitro leaves?
A: We agree with the reviewer. We have made the changes to the manuscript.
Q: Number per which area?
A: We appreciate your comment. We have made the changes to the manuscript.
Q: Please focus the discussion of YOUR results in comparision with literature and avoid too general statements or speculative meanings not based on the presented results
Q: This can be also due to your selection process for the somatic embryos, they could be already very homogenous, but this is not described in your paper. Please prove this statement by adding quantitative data.
A: We appreciate your comment. We refer to the advantages of the SETISTM system; we deposit embryos in the cotyledon stage for their conversion to plants.
Q: But in case of C. canephora you started from zygotic embryos and this should be also taken into account
A: We appreciate your comment. We have made the changes to the manuscript.
Q: Comparing results with these genotypes produced in another system are missing!
A: We appreciate your comment. We have made the changes to the manuscript.
Q: Do you have data from your material? This is a very general publication you cite.
A: We appreciate your comment. We have made the changes to the manuscript.
Q: This is a very general text and I miss your data compared to others. Give quantitative data, please. The pictures do not prived sufficient information and moreover there is no "control" with another system.
A: We agree with the reviewer. We have made the changes to the manuscript.
Q: Data not shown, Line 162. Why??? if this is so important for your discussion.
A: The immersion results from every six hours only produced vitrified seedlings with no more than one centimeter of growth. In contrast, the immersion times of every 24 h did not allow the embryos to become seedlings. The immersion every twelve hours was the one that gave the results shown in the manuscript. The manuscript experiment was repeated twice, and while the manuscript was under evaluation, it was repeated with the same result. This is the reason why the term data not shown was used..
Q: How did you select the embryos, which criteria, which homogenity?
A: We agree with the reviewer. We have made the changes to the manuscript.
Q: Which leaf?. How many leaves?, Number of stomata per which area?
A: We agree with the reviewer. We have made the changes to the manuscript.
Q: Why these data are not shown?
A: We appreciate your comment. We have made the changes to the manuscript.
Q: There was only statistics for acclimatization stage?
A: We appreciate your comment. We have made the changes to the manuscript.

Reviewer 2 Report (Previous Reviewer 1)
The authors give some responses and I feel they are reasonable, and I don't have further questions.
Author Response
Dear reviewer, we appreciate your comments and suggestions on our manuscript. This significantly improved our manuscript.

Reviewer 3 Report (New Reviewer)
Article "In vitro conversion development of Coffea spp. somatic embryos in SETISTM bioreactor a temporary immersion system" by
authors Hugo A. Méndez-Hernández, Rosa M. Galaz-Ávalos, Ana O. Quintana-Escobar, Rodolfo Pech-Hoil, Ana M. Collí-Rodríguez, Itzamná Q. Salas-Peraza, Víctor M. Loyola-Vargas assesses the potential of biotech platforms that use stimulation of regeneration when plants are immersed.
The article is framed according to the rules, contains the necessary sections.
The work is interesting and can find practical application.
There are some remarks about the use of terms. So in the abstract, the authors use the term totipotency, while at present all types of meristematic tissues and tissues with a similar potential have been identified that can lead to regeneration, so this statement does not make sense, and the application of the term totipotency to all leaf cells, to put it mildly, is bewildering , as a question resolved in the last century.
Another minor note, the poor resolution of figure 7.
In general, this methodological article is interesting and can be published with a careful replacement of terms.
It is also necessary to expand the conclusion with a specification of perspectives.
Author Response
Reviewer 3
Article "In vitro conversion development of Coffea spp. somatic embryos in SETISTM bioreactor a temporary immersion system" by
authors Hugo A. Méndez-Hernández, Rosa M. Galaz-Ávalos, Ana O. Quintana-Escobar, Rodolfo Pech-Hoil, Ana M. Collí-Rodríguez, Itzamná Q. Salas-Peraza, Víctor M. Loyola-Vargas assesses the potential of biotech platforms that use stimulation of regeneration when plants are immersed.
The article is framed according to the rules, contains the necessary sections.
The work is interesting and can find practical application.
Q:There are some remarks about the use of terms. So in the abstract, the authors use the term totipotency, while at present all types of meristematic tissues and tissues with a similar potential have been identified that can lead to regeneration, so this statement does not make sense, and the application of the term totipotency to all leaf cells, to put it mildly, is bewildering , as a question resolved in the last century.
A: We agree with the reviewer. We have made the changes to the manuscript.
Q: Another minor note, the poor resolution of figure 7. (Hay que cambiar la figura 7 a más resolución)
A: We changed the figure for a better-quality one.
In general, this methodological article is interesting and can be published with a careful replacement of terms.
Q: It is also necessary to expand the conclusion with a specification of perspectives.
A: We agree with the reviewer. We have made the changes to the manuscript.

Round 2
Reviewer 1 Report (Previous Reviewer 2)
I see improvements of the manuscript. However:
-Not all corrections are done: specie is species
- I do have problems with the term massive propagation because I know only the term mass propagation.
- Hypothesis is still missing.
- Fig 1. … (E) Embryos are in different stages of development (globular, heart, torpedo and Cotyledonary, from left to right).
-line 164 Unclear sentence: A better response was not observed in immersion of one min every 6 and 24 h (Data not shown).
-Make this caption more precise: Figure 4. Diagram of the culture of C. arabica in temporary immersion bioreactors. As a first step, somatic embryos are obtained as described above. leaf explants of in vitro plantlets were obtained. As a first step, leaf explants of in vitro plantlets were obtained. After 45 days, the embryogenic callus was obtained, followed by embryos at different stages of development. The embryos in cotyledonary stage? were deposited in the container of the bioreactors until their conversion to plants
-In 2.4 not corrected e.g. 186.16 mg per ??? (per vessel or what?).
If there is no statistical proved difference you cannot say highest and lowest fresh mass. I have doubts that these differences are worth to be stressed.
-Table 1 is not corrected! Although you mentioned this in your comments.
-Line 241: …rates above 90% . Not answered! The intention was not to have larger pictures, the intention was to have quantitative data.
-Line 312. “The plants’ homogeneity.. „ not data only pictures presented
-Description of statistical is still to general.
Author Response
Reviewer 1
I see improvements of the manuscript. However:
-Not all corrections are done: specie is species
- We apologize for the mistake. We reviewed the entire manuscript and corrected the error.
- I do have problems with the term massive propagation because I know only the term mass propagation.
- We changed the term massive propagation to mass propagation through the manuscript.
- Hypothesis is still missing.
In this case, we respectfully defer to the reviewer.
Not all publications have a hypothesis. A great variety of scientific work does not require a hypothesis, such as the development of a technique, the purification of an enzyme, the characterization of a protein, etc. In our case, it is about developing a methodology to carry out the scale-up of a crop. We have not included a hypothesis since we are not answering a biological question.
- Fig 1. … (E) Embryos are in different stages of development (globular, heart, torpedo and Cotyledonary, from left to right).
- We incorporate the reviewer's suggestion in the legend of Figure 1.
-line 164 Unclear sentence: A better response was not observed in immersion of one min every 6 and 24 h (Data not shown).
- We apologize for the error. The sentence was removed since the following sentence provides the same information.
-Make this caption more precise: Figure 4. Diagram of the culture of C. arabica in temporary immersion bioreactors. As a first step, somatic embryos are obtained as described above. leaf explants of in vitro plantlets were obtained. As a first step, leaf explants of in vitro plantlets were obtained. After 45 days, the embryogenic callus was obtained, followed by embryos at different stages of development. The embryos in cotyledonary stage? were deposited in the container of the bioreactors until their conversion to plants
- We incorporate the reviewer's suggestion in the legend of Figure 4.
-In 2.4 not corrected e.g. 186.16 mg per ??? (per vessel or what?).
- We incorporate the correct units.
If there is no statistical proved difference you cannot say highest and lowest fresh mass. I have doubts that these differences are worth to be stressed.
- In this respect, we respectfully differ from the reviewer. The measured parameters are better in one species than the other; that is a fact. This does not mean, of course, that statistically, there are differences. We have incorporated the following sentence into the text to highlight that although three parameters are higher in C. canephora, they are not statistically significant.
“Although it seems that C. canephora responds slightly better than C. arabica in the bioreactors, the differences are not statistically significant following all the parameters evaluated.”
-Table 1 is not corrected! Although you mentioned this in your comments.
- We apologize for the mistake. We corrected the error.
-Line 241: …rates above 90% . Not answered! The intention was not to have larger pictures, the intention was to have quantitative data.
- Quantitative data are found in Figure 7. We have changed the last sentence of the paragraph to break down the values for the substrate combinations.
-Line 312. “The plants’ homogeneity.. „ not data only pictures presented
- We have decided to delete the sentence as all the material for the homogeneity claim is visual.
-Description of statistical is still to general.
In this case, we respectfully defer to the reviewer.
We decided that comparing data from the in vitro versus in vivo condition was not objective. That is, statistically, we only compared by species of each condition: C. canephora vs. C. arabica in vitro and C. canephora vs C. arabica in vivo. For this reason, the T-student test was carried out, which is adequate to determine whether there is a significant difference between the TWO groups compared.
Both in books and statistics forums, it can be seen that we are using the correct statistical test. An example of this can be found in the following link: https://www.jmp.com/en_is/statistics-knowledge-portal/t-test/two-sample-t-test.html

This manuscript is a resubmission of an earlier submission. The following is a list of the peer review reports and author responses from that submission.
Round 1
Reviewer 1 Report
There is a lack of quantitative data with statistical analysis for evaluating the somatic embryo development of different coffee species in the bioreactor, such as embryogenesis rate, growth, and development of somatic embryos, i.e. number of somatic embryos per treatment (callus or explant), fresh/dry weight of somatic embryos, etc. Also, histology evidence for proving the events of somatic embryogenesis is lacking. Although the authors provide some perfect photos of plant materials, the scientific part of this study is very poor. The data in table 1 need to be analyzed by ANOVA and then give the significant difference between means.
Reviewer 2 Report
Making the maturation stage more efficient is necessary for scaling up SE systems. For this reason, the paper provides interesting information. However, the paper needs careful revision. The text could be shorter and much clearer on the one hand by avoiding redundancy, on the other hand by including important data.
The title could be more precise because TIS, moreover a special TIS, is used only for embryo maturation and conversion not for embryo development at all. Also, the bioreactor should be named here.
In the end of introduction, clear aims or hypothesis should be formulated to understand the scientific value of this work.
Please revise Material and Methods regarding the media used. This is really confusing now. Use always the same term to avoid misunderstanding. Maybe giving all information regarding media in a table would be better for understanding.
In case the same procedure and medium was used for both species as for disinfection and cultivation of zygotic embryo, please give this information only ones. Add also how long the embryos were on this medium and in which stage they were transferred to the pre-conditioning medium.
All information regarding plant material, working procedures, media, growing condition should be given in Material and Methods. Some of this information is so far part of the results.
Most pictures are too small and do not have a size bar.
In Fig 3 and 4 all information of the entire process regarding media, duration, explants could be given. Please note that TIS is only a part of this figures.
Information regarding conversion rate you discuss extensively but there is no data in the results!
Discussion is to general and scientific value is low because of missing hypothesis. Discuss your clearly presented data related to literature. E.g., for conversion rate and homogeneity of plantlets as well as for effect of immersion frequency there are no data presented.
More remarks are annotated to the manuscript.
